# Consistency-Guided Reverse Sampling for General Linear Inverse Problems

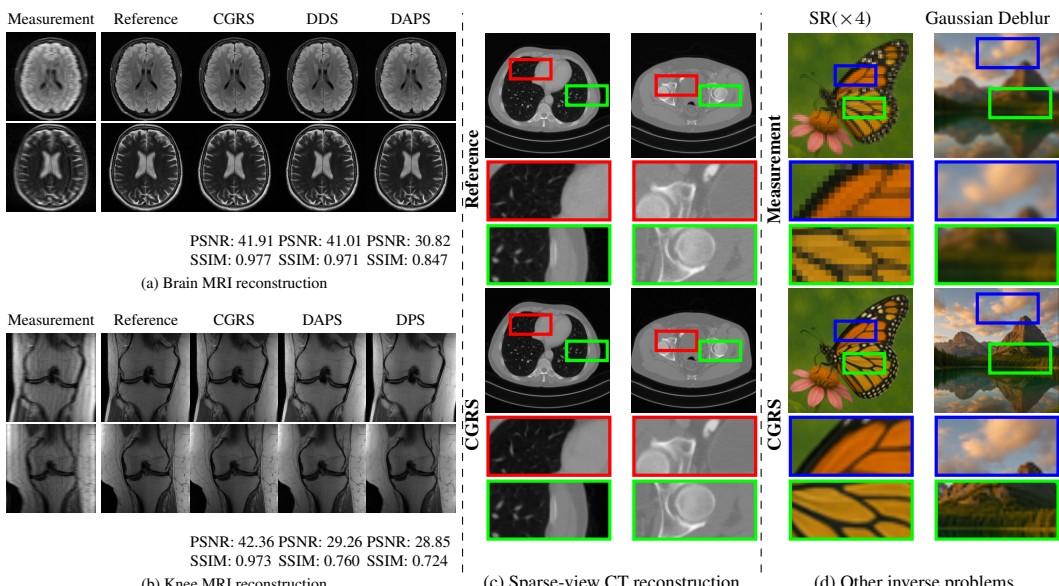

Figure 1: Overview of Consistency-Guided Reverse Sampling (CGRS). Our method introduces a flexible framework for solving a wide range of inverse problems by incorporating consistency constraints into the reverse sampling process. In (a)(b), we show visual comparison on brain and knee MRI reconstruction tasks under highly undersampled conditions, highlighting the ability of our CGRS to preserve anatomical structures and fine details. In (c), we present the performance of our CGRS for sparse-view CT reconstruction, demonstrating improved edge sharpness and reduced artifacts. In (d), we illustrate the performance of our CGRS for classical natural image restoration tasks including super-resolution and Gaussian deblurring on ImageNet, showing strong generalization across different inverse problems.

## Abstract

Diffusion models have recently demonstrated strong potential in solving inverse problems by leveraging the priors of learned data. However, the standard reverse sampling process often struggles to correct errors in the early stages, leading to suboptimal reconstructions, particularly in ill-posed or underdetermined settings. To overcome this issue, we present Consistency-Guided Reverse Sampling (CGRS), a novel framework that integrates measurement-consistency optimization into each reverse diffusion step. CGRS refines the intermediate denoised estimates by solving a linear least-squares problem, thereby improving consistency with the observed measurements. This correction mechanism mitigates error accumulation along the sampling trajectory and enhances overall reconstruction fidelity. Furthermore, CGRS naturally supports flexible acceleration by allowing the reverse process to start from a coarse optimization-based reconstruction, effectively reducing the number of reverse steps with negligible degradation in reconstruction quality. Experimental results in various linear inverse problems

demonstrate that CGRS consistently yields superior performance. Our code is available at https://anonymous.4open.science/r/diffusion-inverse-sampling-00D1.

# 1 INTRODUCTION

Inverse problems exist in many scientific and engineering fields, including computational imaging (Boys et al., 2023; Afonso et al., 2010; Sun & Simon, 2021), medical imaging Suetens (2017); Ravishankar et al. (2019); Sun et al. (2023), and remote sensing (Liu et al., 2021; 2022b). These problems aim to recover an unknown signal $\boldsymbol{x}_0$ from noisy and incomplete measurements $\boldsymbol{y} = \mathcal{A}(\boldsymbol{x}_0) + \boldsymbol{n}$. However, due to the inherent ill-posedness and noise in the forward process, the solution is often ambiguous and unstable.

Recent advances in diffusion models (DMs) (Ho et al., 2020; Song et al., 2020b; Dhariwal & Nichol, 2021; Rombach et al., 2022) have enabled strong generative capabilities by learning complex data distributions via a noise-to-data transformation defined by stochastic differential equations (SDEs). When used as priors in Bayesian inverse problems (Meng & Kabashima, 2022; Chung et al., 2022a; Boys et al., 2023), DMs allow sampling from the posterior distribution by modifying the reverse SDE to include measurement information.

Several works have adopted this idea to solve inverse problems through conditional sampling (Chung et al., 2022a; Song et al., 2023; Chung et al., 2022b), replacing the unconditional score with a conditional one. Despite their effectiveness, existing methods often require hundreds of denoising steps and still suffer from accumulated error and weak measurement consistency.

To address these challenges, we propose *Consistency-Guided Reverse Sampling (CGRS)*, a novel framework that integrates measurement-consistency optimization into each reverse diffusion step. Instead of relying solely on score-based updates, CGRS explicitly enforces consistency with the measurement by projecting intermediate samples back onto the feasible space defined by the forward model. Moreover, we introduce an efficient sampling variant, Fast CGRS, which leverages a coarse optimization-based reconstruction to initialize the reverse process from a lower noise level. This significantly reduces the number of required diffusion steps while maintaining high-quality reconstruction.

We validate CGRS across a range of linear inverse problems, including MRI reconstruction, sparse-view CT reconstruction, and classical image restoration tasks. Experimental results demonstrate that CGRS consistently improves reconstruction fidelity and efficiency compared to state-of-the-art methods, highlighting its robustness and generalizability.

# 2 PRELIMINARIES

## 2.1 DIFFUSION MODELS

Assume that we have a $d$-dimensional random variable $\boldsymbol{x}_0 \in \mathbb{R}^d$ with an unknown distribution $q_0(\boldsymbol{x}_0)$. Diffusion models (DMs) (Ho et al., 2020; Song et al., 2020b; Kingma et al., 2021; Dhariwal & Nichol, 2021; Rombach et al., 2022) usually define a forward process $\{\boldsymbol{x}_t\}_{t \in [0,T]}$ with $T > 0$ starting with $\boldsymbol{x}_0$, such that for any $t \in [0, T]$, the distribution of $\boldsymbol{x}_t$ conditioned on $\boldsymbol{x}_0$ satisfies

$$q_{0t}(\boldsymbol{x}_t \mid \boldsymbol{x}_0) = \mathcal{N}(\boldsymbol{x}_t \mid \sqrt{\bar{\alpha}_t}\boldsymbol{x}_0, (1 - \bar{\alpha}_t)\boldsymbol{I}). \tag{1}$$

The forward process can usually be represented as $\boldsymbol{x}_t = \sqrt{\bar{\alpha}_t}\boldsymbol{x}_0 + \sqrt{1 - \bar{\alpha}_t}\boldsymbol{\epsilon}$ ($\boldsymbol{\epsilon} \sim \mathcal{N}(0, \boldsymbol{I})$), where $\alpha_t \in \mathbb{R}^+$ is a differentiable function of $t$ with bounded derivative, $\bar{\alpha}_t := \prod_{s=1}^t \alpha_s$, and the choice for $\alpha_t$ is referred to as the noise schedule of a DM. (Kingma et al., 2021) have proved that the following stochastic differential equation (SDE) has the same transition distribution $q_{0t}(\boldsymbol{x}_t \mid \boldsymbol{x}_0)$ as in Equation 1 for any $t \in [0, T]$:

$$d\boldsymbol{x}_t = f(\boldsymbol{x}_t, t)\mathrm{d}t + g(t)\mathrm{d}\boldsymbol{w}_t, \qquad \boldsymbol{x}_0 \sim q_0(\boldsymbol{x}_0), \tag{2}$$

where $\boldsymbol{w}_t \in \mathbb{R}^d$ is the standard Wiener process, $f : \mathbb{R}^d \to \mathbb{R}^d$ and $g : \mathbb{R} \to \mathbb{R}$ are the drift and diffusion coefficient that can take different functional forms (e.g., Variance-Preserving SDEs (VP-SDEs) and Variance-Exploding SDEs (VE-SDEs) in (Song et al., 2020b)).

Under some regularity conditions, (Song et al., 2020b) show that the forward process in Equation 2 has an equivalent reverse process from time $T$ to 0, starting with the marginal distribution $q_T(\boldsymbol{x}_T)$:

$$\mathrm{d}\boldsymbol{x}_t = [f(\boldsymbol{x}_t, t) - g^2(t)\nabla_{\boldsymbol{x}_t} \log q_t(\boldsymbol{x}_t)]\mathrm{d}t + g(t)\mathrm{d}\boldsymbol{w}_t,$$
$$\boldsymbol{x}_T \sim q_T(\boldsymbol{x}_T), \tag{3}$$

where $\boldsymbol{w}_t \in \mathbb{R}^d$ is the standard Wiener process in the reverse process. The only unknown term in Equation 3 is the score function $\nabla_{\boldsymbol{x}_t} \log q_t(\boldsymbol{x}_t)$ at each time $t$. In practice, DMs use a neural network $\boldsymbol{\epsilon}_\theta(\boldsymbol{x}_t, t)$ $(\boldsymbol{s}_\theta(\boldsymbol{x}_t, t))$ parameterized by $\theta$ to estimate the scaled score function: $-\sqrt{1 - \bar{\alpha}_t}\nabla_{\boldsymbol{x}_t} \log q_t(\boldsymbol{x}_t)$ $(\nabla_{\boldsymbol{x}_t} \log q_t(\boldsymbol{x}_t))$. The parameter $\theta$ is optimized by minimizing the following objective (Ho et al., 2020; Song et al., 2020b):

$$\min_\theta \mathbb{E}_{q_0(\boldsymbol{x}_0), \boldsymbol{\epsilon}, t} \left[ \omega(t) \| \boldsymbol{\epsilon}_\theta(\boldsymbol{x}_t, t) - \boldsymbol{\epsilon} \|_2^2 \right], \tag{4}$$

or

$$\min_\theta \mathbb{E}_{q_t(\boldsymbol{x}_t), t} \left[ \omega(t) \| \boldsymbol{s}_\theta(\boldsymbol{x}_t, t) + \sqrt{1 - \bar{\alpha}_t}\nabla_{\boldsymbol{x}_t} \log q_t(\boldsymbol{x}_t) \|_2^2 \right], \tag{5}$$

where $\boldsymbol{x}_t \sim q_t(\boldsymbol{x}_t), \boldsymbol{\epsilon} \sim \mathcal{N}(\boldsymbol{0}, \boldsymbol{I}), t \sim \mathcal{U}([0, 1])$, and $\omega(t) > 0$ is a weighting function. Once the parameterized scaled score function $\boldsymbol{\epsilon}\theta$ is obtained, we can approximate the reverse diffusion process in Equation 3 and simulate it using a sampling algorithm. A common choice is the DDPM sampling scheme (Ho et al., 2020), which updates the sample as follows:

$$\boldsymbol{x}_{t-1} = \frac{1}{\sqrt{\alpha_t}}[\boldsymbol{x}_t - \frac{1 - \alpha_t}{\sqrt{1 - \bar{\alpha}_t}}\boldsymbol{\epsilon}_\theta(\boldsymbol{x}_t, t)] + \sqrt{1 - \bar{\alpha}_t}\boldsymbol{z}, \quad \boldsymbol{z} \sim \mathcal{N}(\boldsymbol{0}, \boldsymbol{I}), \tag{6}$$

where $\alpha_t$ is the noise schedule and $\bar{\alpha}_t = \prod_{s=1}^t \alpha_s$.

In practice, the reverse process is not limited to the DDPM update rule, and other numerical solvers (e.g., DDIM (Song et al., 2020a), higher-order ODE/SDE solvers (Lu et al., 2022; Liu et al., 2022a; Zheng et al., 2023)) can also be adopted. More generally, the discrete-time reverse diffusion step can be expressed as

$$\boldsymbol{x}_{t-1} = h(\boldsymbol{x}_t, t; \boldsymbol{\epsilon}_\theta) + \boldsymbol{w}_t, \qquad \boldsymbol{w}_t \sim \mathcal{N}(\boldsymbol{0}, \Sigma_{\boldsymbol{w}}), \tag{7}$$

where $h : \mathbb{R}^d \to \mathbb{R}^d$ is a deterministic update function parameterized by the score function, and $\boldsymbol{w}_t$ is the injected Gaussian noise with covariance $\Sigma_{\boldsymbol{w}}$. Equation 7 thus characterizes the general non-linear dynamics of a discrete-time diffusion sampler.

This general formulation also makes it straightforward to incorporate additional constraints—such as enforcing measurement consistency—into the sampling trajectory, which forms the basis of our proposed Consistency-Guided Reverse Sampling (CGRS) method.

## 2.2 DIFFUSION MODELS FOR SOLVING INVERSE PROBLEMS

Inverse problems, arising from a wide range of applications in various domains, including computational imaging, medical imaging, and remote sensing, among others, aim to recover unknown signals from partial measurements with potential noise. The forward process is usually lossy, resulting in an ill-posed signal recovery task where a unique solution does not exist. In general, a forward model takes the form of

$$\boldsymbol{y} = \mathcal{A}(\boldsymbol{x}_0) + \boldsymbol{n}, \tag{8}$$

where $\mathcal{A}(\cdot) : \mathbb{R}^d \to \mathbb{R}^m$ is the measurement function, $\boldsymbol{x}_0$ represents the original data, $\boldsymbol{y} \in \mathbb{R}^m$ is the observed measurement, and $\boldsymbol{n}$ symbolizes the noise in the measurement process, often modeled as $\boldsymbol{n} \sim \mathcal{N}(\boldsymbol{0}, \beta_{\boldsymbol{y}}^2 \boldsymbol{I})$.

In the Bayesian framework, given measurements $\boldsymbol{y} \in \mathbb{R}^m$ from the forward model Equation 8, one utilizes $q(\boldsymbol{x})$ as the prior, and samples from the posterior distribution $q(\boldsymbol{x} \mid \boldsymbol{y})$, where the relationship is formally established based on the Bayes' rule: $q(\boldsymbol{x} \mid \boldsymbol{y}) \propto q(\boldsymbol{y} \mid \boldsymbol{x})q(\boldsymbol{x})$. Leveraging the diffusion model as the prior, we can solve inverse problems by replacing the score function in Equation 3 with the conditional score function $\nabla_{\boldsymbol{x}_t} \log q_t(\boldsymbol{x}_t \mid \boldsymbol{y})$:

$$\mathrm{d}\boldsymbol{x}_t = \left[f(\boldsymbol{x}_t, t) - g(t)^2 \nabla_{\boldsymbol{x}_t} \log q_t(\boldsymbol{x}_t \mid \boldsymbol{y})\right]\mathrm{d}t + g(t)\mathrm{d}\boldsymbol{w}_t, \tag{9}$$

where, according to Bayes' rule, we can get:

$$\nabla_{\boldsymbol{x}_t} \log q_t(\boldsymbol{x}_t \mid \boldsymbol{y}) = \nabla_{\boldsymbol{x}_t} \log q_t(\boldsymbol{x}_t) + \nabla_{\boldsymbol{x}_t} \log q_t(\boldsymbol{y} \mid \boldsymbol{x}_t). \tag{10}$$

Then, by integrating Equation 10, Equation 9 can generally be written as

$$
\begin{aligned}
\mathrm{d}\boldsymbol{x}_t = [f(\boldsymbol{x}_t, t) - g(t)^2 [\nabla_{\boldsymbol{x}_t} \log q_t(\boldsymbol{x}_t) \\
+ \nabla_{\boldsymbol{x}_t} \log q_t(\boldsymbol{y} \mid \boldsymbol{x}_t)]]\mathrm{d}t + g(t)\mathrm{d}\boldsymbol{w}_t.
\end{aligned}
\tag{11}
$$

Many papers, including classifier-guidance diffusion (Song et al., 2020b; Dhariwal & Nichol, 2021), DPS (Chung et al., 2022a), ΠGDM (Song et al., 2023), MCG (Chung et al., 2022b), MPGD (He et al., 2023), FreeDoM (Yu et al., 2023), DSG (Yang et al., 2024), attempt to leverage pre-trained diffusion models to solve Equation 11 for various inverse problems.

## 3 METHOD

### 3.1 CONSISTENCY-GUIDED REVERSE SAMPLING

---

**Algorithm 1** Consistency-Guided Reverse Sampling (CGRS)

---

**Require:** Model $\boldsymbol{\epsilon}_\theta$, Measurement $\boldsymbol{y}$, Measurement function $\mathcal{A}$, Noise schedule $\{\alpha_t\}_{t=1}^T$.

1: Sample $\boldsymbol{x}_T \sim \mathcal{N}(0, (1 - \bar{\alpha}_T)\boldsymbol{I})$            ▷Initial noise vector
2: **for** $t = T, T-1, \ldots, 1$ **do**
3:    $\hat{\boldsymbol{\epsilon}}_t = \boldsymbol{\epsilon}_\theta(\boldsymbol{x}_t, t)$                ▷Compute the score
4:    $\hat{\boldsymbol{x}}_0(\boldsymbol{x}_t) = \frac{1}{\sqrt{\bar{\alpha}_t}}\left(\boldsymbol{x}_t - \sqrt{1 - \bar{\alpha}_t}\hat{\boldsymbol{\epsilon}}_t\right)$     ▷Predict $\hat{\boldsymbol{x}}_0$ using Tweedie's formula
5:    $\hat{\boldsymbol{x}}_0(\boldsymbol{x}_t, \boldsymbol{y}) \in \arg\min_{\hat{\boldsymbol{x}}_0} \|\boldsymbol{y} - \mathcal{A}(\hat{\boldsymbol{x}}_0(\boldsymbol{x}_t))\|^2,$     ▷Solve with initial point $\hat{\boldsymbol{x}}_0(\boldsymbol{x}_t)$
6:    Sample $\boldsymbol{x}_{t-1} \sim \mathcal{N}(\sqrt{\bar{\alpha}_{t-1}}\hat{\boldsymbol{x}}_0(\boldsymbol{x}_t, \boldsymbol{y}), (1 - \bar{\alpha}_{t-1})\boldsymbol{I})$     ▷Sample $\boldsymbol{x}_{t-1}$
7: **end for**
8: **return** $\boldsymbol{x}_0$

---

When applying diffusion models (DMs) to inverse problems, Equation 7 can be reformulated as

$$
\boldsymbol{x}_{t-1} = h(\boldsymbol{x}_t, \boldsymbol{y}, t; \boldsymbol{\epsilon}_\theta) + \boldsymbol{w}_t, \qquad \boldsymbol{w}_t \sim \mathcal{N}(\boldsymbol{0}, \Sigma_{\boldsymbol{w}}),
\tag{12}
$$

which corresponds to drawing samples from the conditional distribution $q_t(\boldsymbol{x}_{t-1} \mid \boldsymbol{x}_t, \boldsymbol{y})$.

For a reverse step $t \to t-1$ using the Euler–Maruyama discretization, a common and accurate strategy is to introduce $\boldsymbol{x}_0$ as a latent variable and approximate the update through a plug-in posterior. This procedure starts from the identity

$$
q_t(\boldsymbol{x}_{t-1} \mid \boldsymbol{x}_t, \boldsymbol{y}) = \int q_t(\boldsymbol{x}_{t-1} \mid \boldsymbol{x}_t, \boldsymbol{x}_0) q_t(\boldsymbol{x}_0 \mid \boldsymbol{x}_t, \boldsymbol{y}) \, \mathrm{d}\boldsymbol{x}_0.
\tag{13}
$$

In practice, direct evaluation of this integral is generally intractable. We first examine the posterior distribution $q_t(\boldsymbol{x}_0 \mid \boldsymbol{x}_t, \boldsymbol{y})$. By Bayes' rule, it can be expressed as

$$
q_t(\boldsymbol{x}_0 \mid \boldsymbol{x}_t, \boldsymbol{y}) \propto p(\boldsymbol{y} \mid \boldsymbol{x}_0) q_t(\boldsymbol{x}_0 \mid \boldsymbol{x}_t).
\tag{14}
$$

Assuming a Gaussian likelihood, we obtain

$$
p(\boldsymbol{y} \mid \boldsymbol{x}_0) \propto \exp\left(-\frac{1}{2\beta_{\boldsymbol{y}}^2} \|\boldsymbol{y} - \mathcal{A}(\boldsymbol{x}_0)\|_2^2\right)
\tag{15}
$$

Prior works on diffusion posterior sampling (Chung et al., 2022a; Song et al., 2023; Chung et al., 2022b) approximate $q_t(\boldsymbol{x}_0 \mid \boldsymbol{x}_t)$ by a Gaussian of the form

$$
q_t(\boldsymbol{x}_0 \mid \boldsymbol{x}_t) \approx \mathcal{N}\left(\boldsymbol{x}_0; \hat{\boldsymbol{x}}_0(\boldsymbol{x}_t), \frac{1 - \bar{\alpha}_t}{\bar{\alpha}_t}\boldsymbol{I}\right).
\tag{16}
$$

Building on Equation 16, the posterior mean can be expressed in closed form, as stated in Proposition 1, by applying Tweedie's formula (Efron, 2011; Kim & Ye, 2021).

**Proposition 1** *For the case of VP-SDE or VE-SDE sampling (Song et al., 2020b), $q_t(\boldsymbol{x}_0 \mid \boldsymbol{x}_t)$ has the unique posterior mean at*

$$\hat{\boldsymbol{x}}_0(\boldsymbol{x}_t) = \mathbb{E}[\boldsymbol{x}_0 \mid \boldsymbol{x}_t] = \frac{1}{\sqrt{\bar{\alpha}_t}}[\boldsymbol{x}_t + (1 - \bar{\alpha}_t)\boldsymbol{s}_\theta(\boldsymbol{x}_t, t)]. \tag{17}$$

*For the case of DDPM sampling (Ho et al., 2020), the formula can be rewritten as*

$$\hat{\boldsymbol{x}}_0(\boldsymbol{x}_t) = \mathbb{E}[\boldsymbol{x}_0 \mid \boldsymbol{x}_t] = \frac{1}{\sqrt{\bar{\alpha}_t}}[\boldsymbol{x}_t - \sqrt{1 - \bar{\alpha}_t}\boldsymbol{\epsilon}_\theta(\boldsymbol{x}_t, t)]. \tag{18}$$

The posterior mean can be computed as the solution of

$$\hat{\boldsymbol{x}}_0(\boldsymbol{x}_t, \boldsymbol{y}) = \arg\min_{\boldsymbol{x}_0} \frac{1}{2\beta_{\boldsymbol{y}}^2}\|\boldsymbol{y} - \mathcal{A}(\boldsymbol{x}_0(\boldsymbol{x}_t))\|_2^2 + \frac{\bar{\alpha}_t}{2(1 - \bar{\alpha}_t)}\|\boldsymbol{x}_0 - \hat{\boldsymbol{x}}_0(\boldsymbol{x}_t)\|_2^2 \tag{19}$$

At late times (small noise), the influence of the prior diminishes and the update reduces to the simplified correction

$$\hat{\boldsymbol{x}}_0(\boldsymbol{x}_t, \boldsymbol{y}) \approx \arg\min_{\hat{\boldsymbol{x}}_0} \|\boldsymbol{y} - \mathcal{A}(\hat{\boldsymbol{x}}_0(\boldsymbol{x}_t))\|^2, \quad \text{initialized at } \hat{\boldsymbol{x}}_0(\boldsymbol{x}_t), \tag{20}$$

which is equivalent to a measurement-corrected refinement of the denoised estimate. We can use many classical approaches to solve this optimization problem, including the Conjugate Gradient (CG) (Nazareth, 2008), the Alternating Direction Method of Multipliers (ADMM) (Gabay & Mercier, 1976), the Trust Region method (Conn et al., 2000).

In unconditional sampling, the posterior distribution $q_t(\boldsymbol{x}_{t-1} \mid \boldsymbol{x}_t, \boldsymbol{x}_0)$ is often employed since the posterior mean combines information from both the noisy sample $\boldsymbol{x}_t$ and the network-based estimate of $\boldsymbol{x}_0$, which is usually represented as

$$q_t(\boldsymbol{x}_{t-1}|\boldsymbol{x}_t, \boldsymbol{x}_0) = \mathcal{N}\left(\tilde{\mu}(\boldsymbol{x}_t, \boldsymbol{x}_0), \tilde{\beta}_t \boldsymbol{I}\right),$$

$$\text{where } \tilde{\mu}_t(\boldsymbol{x}_t, \boldsymbol{x}_0) = \frac{\sqrt{\bar{\alpha}_{t-1}}(1 - \alpha_t)}{1 - \bar{\alpha}_t}\boldsymbol{x}_0 + \frac{\sqrt{\alpha_t}(1 - \bar{\alpha}_{t-1})}{1 - \bar{\alpha}_t}\boldsymbol{x}_t, \tilde{\beta}_t = \frac{1 - \bar{\alpha}_{t-1}}{1 - \bar{\alpha}_t}(1 - \alpha_t). \tag{21}$$

This formulation reduces variance and stabilizes the reverse diffusion trajectory, which is essential when the estimate of $\boldsymbol{x}_0$ is purely inferred from the generative prior.

However, in conditional sampling scenarios, the situation differs significantly. Once the estimate $\hat{\boldsymbol{x}}_0(\boldsymbol{x}_t, \boldsymbol{y})$ has been refined through measurement guidance, it already incorporates the external information $\boldsymbol{y}$. In this case, relying on the bridge distribution can introduce inconsistency, as its mean $\tilde{\mu}(\boldsymbol{x}_t, \boldsymbol{x}_0)$ couples the condition-informed estimate $\hat{\boldsymbol{x}}_t$ with the unconditional latent $\boldsymbol{x}_t$. This mixture may reintroduce artifacts from the unconditional path and dilute the influence of the measurement constraints.

**Proposition 2** *In conditional diffusion sampling, once the estimate $\hat{\boldsymbol{x}}_0$ incorporates measurement information $\boldsymbol{y}$, the posterior distribution $q_t(\boldsymbol{x}_{t-1} \mid \boldsymbol{x}_t, \hat{\boldsymbol{x}}_0)$ can be well approximated by the marginal distribution $q_t(\boldsymbol{x}_{t-1} \mid \hat{\boldsymbol{x}}_0)$.*

As shown in Proposition 2, we use the marginal distribution

$$q_t(\boldsymbol{x}_{t-1} \mid \hat{\boldsymbol{x}}_0) = \mathcal{N}\left(\sqrt{\bar{\alpha}_{t-1}}\hat{\boldsymbol{x}}_0, (1 - \bar{\alpha}_{t-1})I\right), \tag{22}$$

which treats the condition-informed $\hat{\boldsymbol{x}}_0$ as a surrogate ground-truth signal. This design avoids dependence on the potentially inconsistent $\hat{\boldsymbol{x}}_t$, ensuring that each reverse step propagates information aligned with the observation $\boldsymbol{y}$. As a result, the ancestral form often yields more stable and measurement-consistent reconstructions in conditional diffusion sampling. More details can be found in Appendix A.

Overall, the reverse diffusion process can be expanded into a three-step procedure: (1) sampling $\hat{\boldsymbol{x}}_0(\boldsymbol{x}_t) \sim q_t(\boldsymbol{x}_0 \mid \boldsymbol{x}_t)$, (2) solving $\hat{\boldsymbol{x}}_0(\boldsymbol{x}_t, \boldsymbol{y}) \in \arg\min_{\hat{\boldsymbol{x}}_0} \frac{1}{2}\|\boldsymbol{y} - \mathcal{A}(\hat{\boldsymbol{x}}_0)\|_2^2$ with initial point $\hat{\boldsymbol{x}}_0(\boldsymbol{x}_t)$, and (3) sampling $\boldsymbol{x}_{t-1} \sim \mathcal{N}(\sqrt{\bar{\alpha}_{t-1}}\hat{\boldsymbol{x}}_0(\boldsymbol{x}_t, \boldsymbol{y}), (1 - \bar{\alpha}_{t-1})\boldsymbol{I})$. We repeat this process, gradually reducing noise until $\boldsymbol{x}_0$ is sampled. We refer to this process as *consistency-guided reverse sampling*. The full procedure is summarized in Algorithm 1.

---

**Algorithm 2** Fast CGRS

---

**Require:** Model $\boldsymbol{\epsilon}_\theta$, Measurement $\boldsymbol{y}$, Measurement function $\mathcal{A}$, Noise schedule $\{\alpha_t\}_{t=1}^S$.

1: Initialize vector $\hat{\boldsymbol{x}}_0 = \boldsymbol{0}$
2: $\hat{\boldsymbol{x}}_0 \in \arg\min_{\hat{\boldsymbol{x}}_0} \|\boldsymbol{y} - \mathcal{A}(\hat{\boldsymbol{x}}_0)\|^2$                           ▷Initialize $\hat{\boldsymbol{x}}_0$ with a coarse reconstruction
3: $\boldsymbol{x}_S \sim \mathcal{N}(\sqrt{\bar{\alpha}_S}\hat{\boldsymbol{x}}_0, (1 - \bar{\alpha}_S)\boldsymbol{I})$                           ▷Sample $\boldsymbol{x}_S$
4: **for** $t = S, S-1, \ldots, 1$ **do**
5:     $\hat{\boldsymbol{\epsilon}}_t = \boldsymbol{\epsilon}_\theta(\boldsymbol{x}_t, t)$                           ▷Compute the score
6:     $\hat{\boldsymbol{x}}_0(\boldsymbol{x}_t) = \frac{1}{\sqrt{\bar{\alpha}_t}}\left(\boldsymbol{x}_t - \sqrt{1 - \bar{\alpha}_t}\hat{\boldsymbol{\epsilon}}_t\right)$                           ▷Predict $\hat{\boldsymbol{x}}_0$ using Tweedie's formula
7:     $\boldsymbol{x}_0(\boldsymbol{x}_t) \in \arg\min_{\hat{\boldsymbol{x}}_0} \|\boldsymbol{y} - \mathcal{A}(\hat{\boldsymbol{x}}_0(\boldsymbol{x}_t))\|^2,$                           ▷Solve with initial point $\hat{\boldsymbol{x}}_0(\boldsymbol{x}_t)$
8:     Sample $\boldsymbol{x}_{t-1} \sim \mathcal{N}(\sqrt{\bar{\alpha}_{t-1}}\boldsymbol{x}_0(\boldsymbol{x}_t), (1 - \bar{\alpha}_{t-1})\boldsymbol{I})$                           ▷Sample $\boldsymbol{x}_{t-1}$
9: **end for**
10: **return** $\boldsymbol{x}_0$

---

### 3.2 CGRS FOR FAST SAMPLING

Although diffusion models are known for their superior performance and generalisation capacity, their slow inference time is a critical drawback. Over the years, significant efforts have been made to alleviate this issue, including the development of improved ODE/SDE solvers (Kim et al., 2021; Liu et al., 2022a; Lu et al., 2022; Zheng et al., 2023; Lu et al., 2025) and the adoption of recursive distillation approaches (Salimans & Ho, 2022; Meng et al., 2023). However, most studies have focused on unconditional sampling, leaving conditional sampling acceleration rather under-explored.

A method called *Come-Closer-Diffuse-Faster (CCDF)* (Chung et al., 2022c) first perturbs the initial estimate by applying the forward diffusion process up to a selected time step $t_S < T$, where $t_S$ marks the starting point of the reverse diffusion. Notably, this forward diffusion incurs almost no computational cost, as it does not require any neural network evaluations. We found that the *CCDF* ideas can be effectively incorporated into the *CGRS*.

Unlike *CCDF*, which samples $\boldsymbol{x}_S$ at time $t_S$ directly from the measurement $\boldsymbol{y}$ via $\boldsymbol{x}_S \sim \mathcal{N}(\sqrt{\bar{\alpha}_S}\boldsymbol{y}, (1 - \bar{\alpha}_S)\boldsymbol{I})$, our approach replaces $\boldsymbol{y}$ with a coarse reconstruction obtained by solving the optimization problem $\arg\min_{\hat{\boldsymbol{x}}_0} \|\boldsymbol{y} - \mathcal{A}(\hat{\boldsymbol{x}}_0)\|^2$. Specifically, we perform the following steps:

$$\text{Initialize vector } \hat{\boldsymbol{x}}_0 = \boldsymbol{0},$$
$$\hat{\boldsymbol{x}}_0 \in \arg\min_{\hat{\boldsymbol{x}}_0} \|\boldsymbol{y} - \mathcal{A}(\hat{\boldsymbol{x}}_0)\|^2, \tag{23}$$
$$\boldsymbol{x}_S \sim \mathcal{N}(\sqrt{\bar{\alpha}_S}\hat{\boldsymbol{x}}_0, (1 - \bar{\alpha}_S)\boldsymbol{I}).$$

This process usually requires hundreds of iterations, which requires several seconds to complete. The full procedure is summarized in Algorithm 2.

## 4 EXPERIMENTS

### 4.1 EXPERIMENTAL SETUP

We evaluate our method using the pre-trained diffusion models provided by (Chung et al., 2023). Specifically, three pixel-space models are used, each trained on a different dataset: the fastMRI brain dataset (Zbontar et al., 2018), the fastMRI knee dataset (Zbontar et al., 2018), and the AAPM LDCT dataset (Moen et al., 2021).

**Datasets and metrics.** Building on the previously introduced models, we evaluate our method on the fastMRI and AAPM LDCT datasets. Specifically, we use 1,000 images from the fastMRI brain test set, 1,000 images from the fastMRI knee validation set, and 500 images from the AAPM LDCT validation set. Our primary evaluation metrics include peak signal-to-noise ratio (PSNR), structural similarity index measure (SSIM), and Learned Perceptual Image Patch Similarity (LPIPS) (Zhang et al., 2018). For both our method and the baselines, we use the implementations provided by TorchMetrics (Nicki Skafte Detlefsen et al., 2022), with all images normalized to the [0, 1] range.

**Problem setting.** We have the following general measurement model

$$\boldsymbol{y} = \mathcal{M}\mathcal{T}\mathcal{S}(\boldsymbol{x}) = \mathcal{A}(\boldsymbol{x}), \boldsymbol{x} \in \mathbb{R}^d, \boldsymbol{y} \in \mathbb{R}^m, \mathcal{A} \in \mathbb{R}^{d \times m}, \tag{24}$$

where $\mathcal{M}$ is the sub-sampling matrix, $\mathcal{T}$ is the discrete transform matrix (i.e. Fourier, Radon), and $\mathcal{S} = \boldsymbol{I}$ when we have a single-array measurement including CT, and $\mathcal{S} = [\mathcal{S}^{(1)}, \ldots, \mathcal{S}^{(c)}]$ when we have a $c$-coil MR parallel imaging measurement.

**Baselines.** For MRI reconstruction, we compare our CGRS method with several diffusion models. These include DPS (Chung et al., 2022a), DAPS (Zhang et al., 2025), and DDS (Chung et al., 2023), which are designed for general image restoration tasks, as well as ScoreMRI (Chung & Ye, 2022) and CSGM (Jalal et al., 2021), which are specifically tailored for MRI reconstruction using diffusion models. All experiments are conducted using equispaced sub-sampling masks. For CT reconstruction, we benchmark our method against DPS, MCG (Chung et al., 2022b), and DDS. In addition to these general-purpose inverse problem solvers, we also compare with two representative baselines: (1) Filtered Backprojection (FBP), a conventional analytic reconstruction technique, and (2) FBP-UNet (Jin et al., 2017), a supervised deep learning model that refines FBP outputs using a UNet architecture.

## 4.2 MAIN RESULTS

| Task | Method | ×4 Acceleration | | | ×8 Acceleration | | | ×12 Acceleration | | |
|------|--------|-------|-------|--------|-------|-------|--------|-------|-------|--------|
| | | PSNR↑ | SSIM↑ | LPIPS↓ | PSNR↑ | SSIM↑ | LPIPS↓ | PSNR↑ | SSIM↑ | LPIPS↓ |
| Brain-MRI | CSGM(NFE = 2311) | 33.73 | 0.788 | 0.072 | 30.52 | 0.776 | 0.113 | 27.64 | 0.757 | 0.146 |
| | ScoreMRI(NFE = 1000) | 28.46 | 0.719 | 0.128 | 25.79 | 0.693 | 0.142 | 24.33 | 0.681 | 0.157 |
| | DPS(NFE = 1000) | 29.34 | 0.739 | 0.117 | 26.97 | 0.711 | 0.133 | 26.50 | 0.701 | 0.136 |
| | DAPS(NFE = 1000) | 31.69 | 0.829 | 0.058 | 28.30 | 0.778 | 0.094 | 27.11 | 0.753 | 0.108 |
| | DDS(NFE = 1000) | 40.49 | 0.957 | 0.009 | 34.06 | 0.893 | 0.033 | 31.25 | 0.853 | 0.055 |
| | CGRS(NFE = 1000) | **42.51** | **0.972** | **0.006** | **34.44** | **0.906** | **0.029** | **31.37** | **0.859** | **0.051** |
| Knee-MRI | CSGM(NFE = 2311) | 33.24 | 0.774 | 0.118 | 30.26 | 0.752 | 0.148 | 27.18 | 0.741 | 0.183 |
| | ScoreMRI(NFE = 1000) | 27.75 | 0.689 | 0.156 | 25.13 | 0.664 | 0.173 | 23.89 | 0.642 | 0.198 |
| | DPS(NFE = 1000) | 28.92 | 0.729 | 0.146 | 26.81 | 0.684 | 0.166 | 26.26 | 0.666 | 0.169 |
| | DAPS(NFE = 1000) | 30.94 | 0.779 | 0.106 | 28.07 | 0.709 | 0.152 | 26.93 | 0.676 | 0.165 |
| | DDS(NFE = 1000) | 39.67 | 0.945 | 0.017 | 33.99 | 0.868 | 0.053 | 31.13 | 0.814 | 0.085 |
| | CGRS(NFE = 1000) | **41.73** | **0.965** | **0.011** | **34.31** | **0.886** | **0.042** | **31.25** | **0.836** | **0.078** |

Table 1: Quantitative results of MRI reconstruction on fastMRI data. Performance comparison on brain and knee MRI reconstruction with different methods across different acceleration factors (×4, ×8, ×12). NFE: number of function evaluations. The best and second-best results within each type of task are indicated in bold and underlined, respectively.

We present the quantitative results for brain and knee MRI reconstruction tasks under various acceleration settings in Table 1. As shown, our method, CGRS, consistently outperforms existing baselines across all configurations, achieving superior reconstruction accuracy and perceptual quality. Notably, CGRS maintains superior and stable performance for both brain and knee MRI reconstructions, even under high acceleration factors. Our model not only preserves global anatomical structures but also faithfully restores subtle textures and fine-grained details that are often lost by competing methods.

While other approaches tend to suffer from notable performance degradation as the acceleration factor increases—manifesting in blurred structures and perceptual artifacts—CGRS demonstrates exceptional stability. It excels at preserving high-frequency information and ensuring visual consistency across varying levels of measurement sparsity. This consistent performance across different anatomical regions and sampling conditions highlights the versatility and robustness of CGRS in addressing a broad range of MRI inverse problems.

Table 2 summarizes the quantitative results for CT reconstruction under sparse-view settings. As shown, CGRS consistently achieves the best performance across all the evaluation metrics. Compared to traditional baselines such as FBP and its UNet-enhanced variant, CGRS delivers reconstructions with significantly improved image fidelity and structural integrity. Compared to more advanced diffusion models including DPS and MCG, CGRS demonstrates apparent advantages since these methods often struggle to recover fine-grained details or suffer from diminished perceptual quality under sparse measurements. In contrast, CGRS strikes an effective balance between reconstruction accuracy and visual realism, achieving leading results across both pixel-level and perceptual evaluations. Additional visual results are provided in Appendix B to further highlight the effectiveness of CGRS.

| Method | 18 Views | | |
|---|---|---|---|
| | PSNR↑ | SSIM↑ | LPIPS↓ |
| FBP | 14.14 | 0.346 | 0.522 |
| FBP-UNet | 30.24 | 0.878 | 0.042 |
| DPS | 28.56 | 0.820 | 0.059 |
| MCG | 27.47 | 0.778 | 0.116 |
| DDS | 32.04 | 0.912 | 0.018 |
| CGRS | **34.68** | **0.928** | **0.012** |

Table 2: Quantitative evaluation on AAPM data. Performance comparison on CT reconstruction with different methods across a parallel-beam geometry using 18 projection angles equally distributed across 180 degrees using the torch-radon package (Ronchetti, 2020). Bold indicates the best results.

## 4.3 FURTHER RESULTS ON NATURAL IMAGE

| Method | PSNR↑ | SSIM↑ | LPIPS↓ |
|---|---|---|---|
| Diff-OC | 28.22 | 0.832 | 0.365 |
| DDRM | 27.87 | 0.824 | 0.368 |
| DPS | 29.16 | 0.839 | 0.352 |
| DDNM | 28.96 | 0.826 | 0.362 |
| DAPS | 30.03 | 0.869 | 0.344 |
| CGRS | **32.55** | **0.915** | **0.329** |

Table 3: Quantitative evaluation of super-resolution ($\times$4) on FFHQ data.

We further evaluate the effectiveness of CGRS on the FFHQ (Karras et al., 2019) and ImageNet (Deng et al., 2009) datasets for classical inverse problems, including super-resolution and Gaussian deblurring. For the Gaussian deblurring task, we use a kernel of size $61 \times 61$ with a standard deviation of 3. In the super-resolution task, we employ a bicubic resizer. We leverage pre-trained diffusion models: one trained on the FFHQ dataset by (Chung et al., 2022a), and another pre-trained model from (Dhariwal & Nichol, 2021) on the ImageNet dataset. We compare our methods with the following methods: DAPS, DDRM (Kawar et al., 2022), DPS, DDNM (Wang et al., 2022), Diff-OC (Li & Pereira, 2024) for super-resolution and Gaussian deblurring experiments. Table 3 shows that CGRS outperforms the existing diffusion-based methods significantly on FFHQ dataset. We include more qualitative results in Appendix B.

## 4.4 ABLATION STUDY

As illustrated in Figure 2, Fast CGRS achieves significantly higher PSNR than the existing baselines under all sampling budgets. This is primarily attributed to the coarse initialization obtained via solving an optimization problem $\arg\min_{\hat{x}_0} \|y - \mathcal{A}(\hat{x}_0)\|^2$ first, which effectively narrows the sampling space and accelerates convergence. Unlike conventional methods that sample from Gaussian noise, our Fast CGRS method benefits from a structured prior of MR image reconstructed by a fast coarse reconstruction, significantly narrowing the sampling space and accelerating convergence.

| | Super-Resolution ($\times$4) | | |
|---|---|---|---|
| | PSNR↑ | SSIM↑ | LPIPS↓ |
| Fast CGRS (NFE/S = 10) | 26.59 | 0.783 | 0.375 |
| Fast CGRS (NFE/S = 50) | 30.59 | 0.875 | 0.344 |
| Fast CGRS (NFE/S = 100) | 31.25 | 0.886 | 0.342 |
| Fast CGRS (NFE/S = 200) | 31.28 | 0.887 | 0.338 |
| Fast CGRS (NFE/S = 300) | 31.32 | 0.889 | 0.337 |
| CGRS (NFE = 1000) | 32.15 | 0.896 | 0.332 |

Table 4: Quantitative evaluation of different initialization steps on 50 FFHQ images.

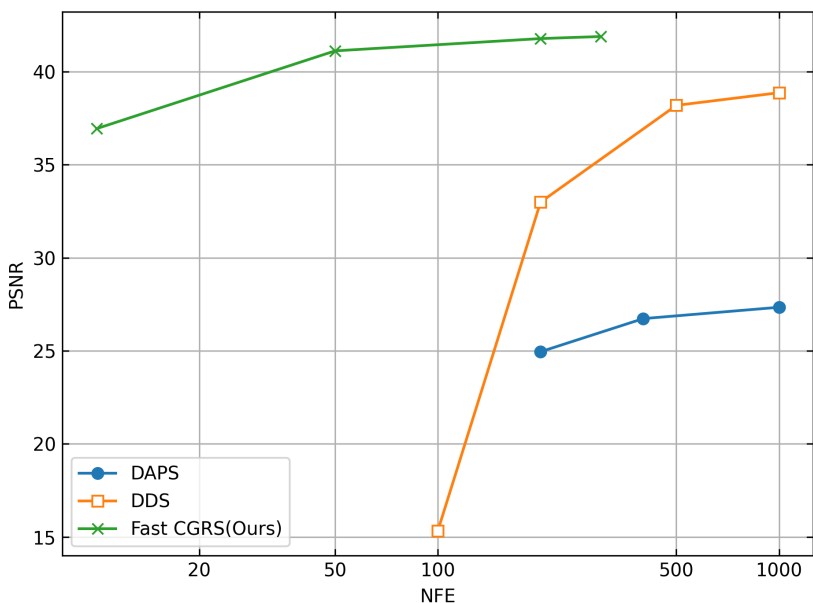

Figure 2: MRI reconstruction evaluated by PSNR vs. NFE (log scale) with equispaced sub-sampling ×4.

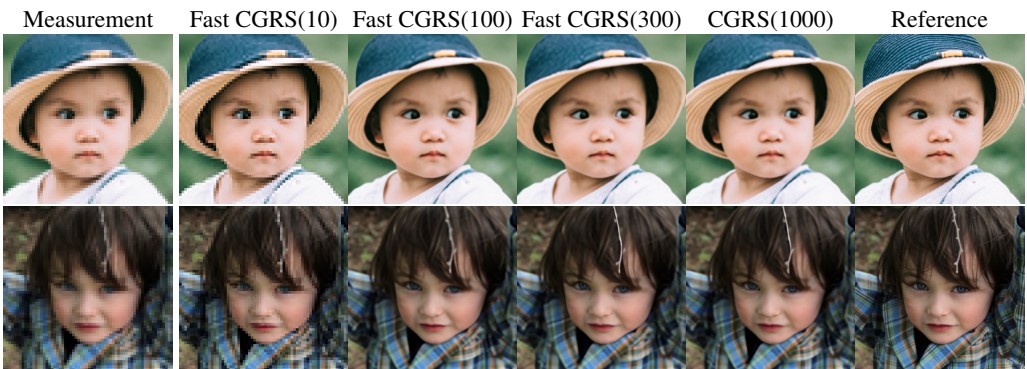

Figure 3: Visualization of different initialization steps on FFHQ dataset.

To further investigate the effect of the initialization procedure, we study the impact of selecting different starting points $S$ for the reverse diffusion process. Table 4 reports the quantitative performance when varying $S$. The results show that larger $S$ values (i.e., starting the reverse process from a later diffusion step) generally improve PSNR and SSIM while reducing LPIPS. However, the improvements plateau beyond a certain point, indicating diminishing returns. This trade-off between speed and reconstruction quality suggests that an intermediate range of NFE (e.g., 100–200) provides a balanced choice in practice, offering both computational efficiency and reliable visual fidelity. The visualized results are shown in Figure 3.

## 5 CONCLUSION

In summary, we propose Consistency-Guided Reverse Sampling (CGRS) for solving general linear inverse problems. Our method enforces measurement consistency at each step of the reverse diffusion sampling trajectory by integrating optimization-based correction, thereby correcting early-stage errors and improving data fidelity. This design also supports flexible acceleration by initializing the reverse process from a coarse optimization-based reconstruction, reducing the number of sampling steps without compromising quality. In the experiments, we demonstrate that CGRS generates samples with improved reconstruction accuracy, stability, and efficiency compared to the existing diffusion-based methods across a wide range of challenging linear inverse problems.

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

# APPENDIX

## A PROOF FOR PROPOSITIONS

**Proposition 3** *(Restated). For the case of VP-SDE or VE-SDE sampling (Song et al., 2020b), $q_t(\boldsymbol{x}_0 \mid \boldsymbol{x}_t)$ has the unique posterior mean at*

$$\hat{\boldsymbol{x}}_0(\boldsymbol{x}_t) = \mathbb{E}[\boldsymbol{x}_0 \mid \boldsymbol{x}_t] = \frac{1}{\sqrt{\bar{\alpha}_t}}[\boldsymbol{x}_t + (1 - \bar{\alpha}_t)\boldsymbol{s}_\theta(\boldsymbol{x}_t, t)]. \tag{25}$$

*For the case of DDPM sampling (Ho et al., 2020), the formula can be rewritten as*

$$\hat{\boldsymbol{x}}_0(\boldsymbol{x}_t) = \mathbb{E}[\boldsymbol{x}_0 \mid \boldsymbol{x}_t] = \frac{1}{\sqrt{\bar{\alpha}_t}}[\boldsymbol{x}_t - \sqrt{1 - \bar{\alpha}_t}\boldsymbol{\epsilon}_\theta(\boldsymbol{x}_t, t)]. \tag{26}$$

**Proof.** Consider the forward process $\boldsymbol{x}_t = \sqrt{\bar{\alpha}_t}\boldsymbol{x}_0 + \sigma_t z$ with $z \sim \mathcal{N}(0, I)$, where $\sigma_t = \sqrt{1 - \bar{\alpha}_t}$. Lettin $\boldsymbol{x} = \sqrt{\bar{\alpha}_t}\boldsymbol{x}_0$, we want to compute

$$\mathbb{E}[\boldsymbol{x} \mid \boldsymbol{x}_t] = \int \boldsymbol{x}\, q_t(\boldsymbol{x} \mid \boldsymbol{x}_t)\, \mathrm{d}\boldsymbol{x}. \tag{27}$$

Using Bayes' rule,

$$\mathbb{E}[\boldsymbol{x} \mid \boldsymbol{x}_t] = \int \boldsymbol{x}\frac{q_t(\boldsymbol{x}_t \mid \boldsymbol{x})q_t(\boldsymbol{x})}{q_t(\boldsymbol{x}_t)}\mathrm{d}\boldsymbol{x} = \frac{1}{q_t(\boldsymbol{x}_t)}\int \boldsymbol{x}\, q_t(\boldsymbol{x}_t \mid \boldsymbol{x})\, q_t(\boldsymbol{x})\, \mathrm{d}\boldsymbol{x}. \tag{28}$$

Since $\boldsymbol{x}_t \mid \boldsymbol{x} \sim \mathcal{N}(\boldsymbol{x}, \sigma_t^2 I)$, we have

$$q_t(\boldsymbol{x}_t \mid \boldsymbol{x}) = \frac{1}{\sqrt{2\pi\sigma_t^2}}\exp\left(-\frac{\|\boldsymbol{x}_t - \boldsymbol{x}\|^2}{2\sigma_t^2}\right). \tag{29}$$

Thus,

$$\mathbb{E}[\boldsymbol{x} \mid \boldsymbol{x}_t] = \frac{1}{q_t(\boldsymbol{x}_t)}\int \boldsymbol{x}\, \frac{1}{\sqrt{2\pi\sigma_t^2}}\exp\left(-\frac{\|\boldsymbol{x}_t - \boldsymbol{x}\|^2}{2\sigma_t^2}\right) q_t(\boldsymbol{x})\, \mathrm{d}\boldsymbol{x}. \tag{30}$$

Multiply numerator and denominator by $\sqrt{\bar{\alpha}_t}$, and rewrite

$$\boldsymbol{x} = \sigma_t^2\frac{\boldsymbol{x} - \boldsymbol{x}_t}{\sigma_t^2} + \boldsymbol{x}_t. \tag{31}$$

So,

$$\mathbb{E}[\boldsymbol{x} \mid \boldsymbol{x}_t] = \frac{1}{q_t(\boldsymbol{x}_t)}\int (\sigma_t^2\frac{\boldsymbol{x} - \boldsymbol{x}_t}{\sigma_t^2} + \boldsymbol{x}_t)\frac{1}{\sqrt{2\pi\sigma_t^2}}\exp\left(-\frac{\|\boldsymbol{x}_t - \boldsymbol{x}\|^2}{2\sigma_t^2}\right) q_t(\boldsymbol{x})\, \mathrm{d}\boldsymbol{x}. \tag{32}$$

This splits into two terms,

$$\begin{aligned}\mathbb{E}[\boldsymbol{x} \mid \boldsymbol{x}_t] &= \frac{1}{q_t(\boldsymbol{x}_t)}\int \sigma_t^2\frac{\boldsymbol{x} - \boldsymbol{x}_t}{\sigma_t^2}\frac{1}{\sqrt{2\pi\sigma_t^2}}\exp\left(-\frac{\|\boldsymbol{x}_t - \boldsymbol{x}\|^2}{2\sigma_t^2}\right) q_t(\boldsymbol{x})\, \mathrm{d}\boldsymbol{x} \\ &+ \frac{1}{q_t(\boldsymbol{x}_t)}\int \boldsymbol{x}_t\frac{1}{\sqrt{2\pi\sigma_t^2}}\exp\left(-\frac{\|\boldsymbol{x}_t - \boldsymbol{x}\|^2}{2\sigma_t^2}\right) q_t(\boldsymbol{x})\, \mathrm{d}\boldsymbol{x}.\end{aligned} \tag{33}$$

Now, notice that the first term is

$$\begin{aligned}&\frac{1}{q_t(\boldsymbol{x}_t)}\int \sigma_t^2\frac{\boldsymbol{x} - \boldsymbol{x}_t}{\sigma_t^2}\frac{1}{\sqrt{2\pi\sigma_t^2}}\exp\left(-\frac{\|\boldsymbol{x}_t - \boldsymbol{x}\|^2}{2\sigma_t^2}\right) q_t(\boldsymbol{x})\, \mathrm{d}\boldsymbol{x} \\ &= \frac{\sigma_t^2}{q_t(\boldsymbol{x}_t)}\int \frac{\mathrm{d}\left[\frac{1}{\sqrt{2\pi\sigma_t^2}}\exp\left(-\frac{\|\boldsymbol{x}_t - \boldsymbol{x}\|^2}{2\sigma_t^2}\right)\right]}{\mathrm{d}\boldsymbol{x}_t}q_t(\boldsymbol{x})\, \mathrm{d}\boldsymbol{x} \\ &= \frac{\sigma_t^2}{q_t(\boldsymbol{x}_t)}\frac{\mathrm{d}}{\mathrm{d}\boldsymbol{x}_t}\int q_t(\boldsymbol{x}_t \mid \boldsymbol{x})\, q_t(\boldsymbol{x})\, \mathrm{d}\boldsymbol{x} \\ &= \frac{\sigma_t^2}{q_t(\boldsymbol{x}_t)}\frac{\mathrm{d}q_t(\boldsymbol{x}_t)}{\mathrm{d}\boldsymbol{x}_t} \\ &= \sigma_t^2\nabla_{\boldsymbol{x}_t}\log q_t(\boldsymbol{x}_t),\end{aligned} \tag{34}$$

and the second term is

$$\frac{1}{q_t(\boldsymbol{x}_t)} \int \boldsymbol{x}_t \frac{1}{\sqrt{2\pi\sigma_t^2}} \exp\left(-\frac{\|\boldsymbol{x}_t - \boldsymbol{x}\|^2}{2\sigma_t^2}\right) q_t(\boldsymbol{x})\, \mathrm{d}\boldsymbol{x}$$
$$= \frac{1}{q_t(\boldsymbol{x}_t)} \int \boldsymbol{x}_t q_t(\boldsymbol{x}_t \mid \boldsymbol{x})\, q_t(\boldsymbol{x})\, \mathrm{d}\boldsymbol{x} \qquad (35)$$
$$= \boldsymbol{x}_t$$

Hence,

$$\mathbb{E}[\boldsymbol{x} \mid \boldsymbol{x}_t] = \boldsymbol{x}_t + \sigma_t^2 \nabla_{\boldsymbol{x_t}} \log q_t(\boldsymbol{x_t}) \qquad (36)$$

Finally, we expand $\boldsymbol{x} = \sqrt{\bar{\alpha}_t}\boldsymbol{x}_0$ and invoke the linearity of the expectation to arrive at

$$\mathbb{E}[\boldsymbol{x}_0 \mid \boldsymbol{x}_t] = \frac{1}{\sqrt{\bar{\alpha}_t}}[\boldsymbol{x}_t + \sigma_t^2 \nabla_{\boldsymbol{x_t}} \log q_t(\boldsymbol{x_t})] \qquad (37)$$

In diffusion models, the $\epsilon$-prediction parameterization estimates the injected noise term $\epsilon_\theta$, whereas the score-prediction parameterization estimates the score function $\boldsymbol{s}_\theta$. Their relationship is given by

$$\begin{aligned} \epsilon_\theta &= -\sigma_t \nabla_{\boldsymbol{x_t}} \log q_t(\boldsymbol{x}_t), \\ \boldsymbol{s}_\theta &= \nabla_{\boldsymbol{x_t}} \log q_t(\boldsymbol{x}_t) \end{aligned} \qquad (38)$$

Accordingly, the posterior mean of the clean sample $\boldsymbol{x}_0$ given $\boldsymbol{x}_t$ can be expressed in two equivalent forms:

$$\hat{\boldsymbol{x}}_0(\boldsymbol{x}_t) = \mathbb{E}[\boldsymbol{x}_0 \mid \boldsymbol{x}_t] = \frac{1}{\sqrt{\bar{\alpha}_t}}[\boldsymbol{x}_t + (1 - \bar{\alpha}_t)\boldsymbol{s}_\theta(\boldsymbol{x}_t, t)],$$

$$\hat{\boldsymbol{x}}_0(\boldsymbol{x}_t) = \mathbb{E}[\boldsymbol{x}_0 \mid \boldsymbol{x}_t] = \frac{1}{\sqrt{\bar{\alpha}_t}}[\boldsymbol{x}_t - \sqrt{1 - \bar{\alpha}_t}\epsilon_\theta(\boldsymbol{x}_t, t)].$$

$\square$

**Proposition 4** *(Restated). In conditional diffusion sampling, once the estimate $\hat{\boldsymbol{x}}_0$ incorporates measurement information $\boldsymbol{y}$, the posterior distribution $q(\boldsymbol{x}_{t-1} \mid \boldsymbol{x}_t, \hat{\boldsymbol{x}}_0)$ can be well approximated by the marginal distribution $q(\boldsymbol{x}_{t-1} \mid \hat{\boldsymbol{x}}_0)$.*

**Proof.** In unconditional diffusion sampling, substituting the clean signal $\boldsymbol{x}_0$ into the posterior distribution

$$q_t(\boldsymbol{x}_{t-1} \mid \boldsymbol{x}_t, \boldsymbol{x}_0) = \mathcal{N}\big(\boldsymbol{x}_{t-1}; \tilde{\mu}_t(\boldsymbol{x}_t, \boldsymbol{x}_0), \tilde{\beta}_t I\big), \quad \tilde{\beta}_t = \frac{1 - \bar{\alpha}_{t-1}}{1 - \bar{\alpha}_t}\beta_t, \quad \beta_t = 1 - \alpha_t \qquad (39)$$

and the true reverse posterior mean is

$$\tilde{\mu}_t(\boldsymbol{x}_t, \boldsymbol{x}_0) = \frac{\sqrt{\bar{\alpha}_{t-1}}\beta_t}{1 - \bar{\alpha}_t}\boldsymbol{x}_0 + \frac{\sqrt{\alpha_t}(1 - \bar{\alpha}_{t-1})}{1 - \bar{\alpha}_t}\boldsymbol{x}_t. \qquad (40)$$

Since $\boldsymbol{x}_0$ and $\boldsymbol{x}_t$ come from the same forward process, they are matched. Integrating over $\boldsymbol{x}_0$ gives the exact marginal

$$q_{t-1}(\boldsymbol{x}_{t-1}) \propto \int q(\boldsymbol{x}_{t-1} \mid \boldsymbol{x}_t, \boldsymbol{x}_0)\, q(\boldsymbol{x}_0 \mid \boldsymbol{x}_t)\mathrm{d}\boldsymbol{x}_t. \qquad (41)$$

Thus, the marginal distribution $q_{t-1}(\boldsymbol{x}_{t-1})$ is exact in the unconditional case.

However, in conditional sampling with measurement guidance, when $\boldsymbol{x}_0$ is replaced by the estimator $\hat{\boldsymbol{x}}_0(\boldsymbol{x}_t, \boldsymbol{y})$, the update

$$q_t(\boldsymbol{x}_{t-1} \mid \boldsymbol{x}_t, \hat{\boldsymbol{x}}_0) = \mathcal{N}\big(\boldsymbol{x}_{t-1}; \tilde{\mu}_t(\boldsymbol{x}_t, \hat{\boldsymbol{x}}_0), \tilde{\beta}_t I\big). \qquad (42)$$

Substituting into the bridge posterior yields

$$\tilde{\mu}_t(\boldsymbol{x}_t, \hat{\boldsymbol{x}}_0) = \frac{\sqrt{\bar{\alpha}_{t-1}}\beta_t}{1 - \bar{\alpha}_t}\hat{\boldsymbol{x}}_0 + \frac{\sqrt{\alpha_t}(1 - \bar{\alpha}_{t-1})}{1 - \bar{\alpha}_t}\boldsymbol{x}_t. \qquad (43)$$

Here $\hat{x}_0(x_t, y)$ encodes information from $y$, but the noise term $\epsilon_\theta(x_t, t)$ (hidden inside the dependence on $x_t$) is still unconditional. This mismatch introduces an error in the mean:

$$\Delta_t = \tilde{\mu}_t(x_t, \hat{x}_0) - \tilde{\mu}_t(x_t, x_0). \tag{44}$$

Due to $\hat{x}_0(x_t, y) \approx \arg\min_{\hat{x}_0} \|y - \mathcal{A}(\hat{x}_0)\|^2$, and $\hat{x}_0$ is independent of $x_t$, the marginal distribution $q_{t-1}(x_{t-1})$ approximation discards the dependence on $x_t$

$$q_{t-1}(x_{t-1}) \propto q(x_{t-1} \mid \hat{x}_0)\, q(\hat{x}_0), \tag{45}$$

which depends solely on the quality of $\hat{x}_0$ and avoids the inconsistency term from $x_t$. Consequently, $q(x_{t-1} \mid \hat{x}_0)$ provides a cleaner and measurement-consistent reverse step.

$\square$

## B  ADDITIONAL EXPERIMENTS

### B.1  EXPERIMENTAL RESULTS ON FASTMRI AND AAPM LDCT

Figure 4, 5, 6, provide additional visualization results for Brain MRI, Knee MRI, and CT reconstruction tasks under different undersampling settings. These examples complement the quantitative evaluations in the main text and highlight the ability of CGRS to produce high-fidelity reconstructions with sharp structural details and reduced artifacts across diverse tasks.

### B.2  EXPERIMENTAL RESULTS ON FFHQ AND IMAGENET

| Method | PSNR↑ | SSIM↑ | LPIPS↓ |
|---|---|---|---|
| Diff-OC | 25.24 | 0.738 | 0.374 |
| DDRM | 24.43 | 0.721 | 0.393 |
| DPS | 27.50 | 0.797 | 0.352 |
| DDNM | 27.68 | 0.815 | 0.364 |
| DAPS | 29.94 | 0.827 | 0.355 |
| CGRS | **35.79** | **0.953** | **0.317** |

Table 5: **Quantitative evaluation of Gaussian deblurring on FFHQ**. The best and second-best results within each type of task are indicated by bold and underlined marks, respectively.

| Method | Super-Resolution (×4) | | | Gaussian Deblurring | | |
|---|---|---|---|---|---|---|
| | PSNR↑ | SSIM↑ | LPIPS↓ | PSNR↑ | SSIM↑ | LPIPS↓ |
| Diff-OC | 22.28 | 0.615 | 0.398 | 22.19 | 0.539 | 0.425 |
| DDRM | 20.12 | 0.586 | 0.443 | 20.07 | 0.522 | 0.451 |
| DPS | 23.05 | 0.612 | 0.369 | 22.19 | 0.539 | 0.396 |
| DDNM | 21.54 | 0.597 | 0.436 | 21.37 | 0.527 | 0.446 |
| DAPS | 23.22 | 0.594 | 0.396 | 22.94 | 0.574 | 0.413 |
| CGRS | **23.88** | **0.663** | **0.354** | **25.47** | **0.757** | **0.357** |

Table 6: **Quantitative evaluation on ImageNet**. The best and second-best results within each type of task are indicated by bold and underlined marks, respectively.

Tab 5, 6 provide additional quantitative results on FFHQ and ImageNet datasets, respectively. CGRS outperforms existing diffusion-based baselines. Figure 7, 8, 9, 10 provide more visualization results for super-resolution and Gaussian deblurring tasks.

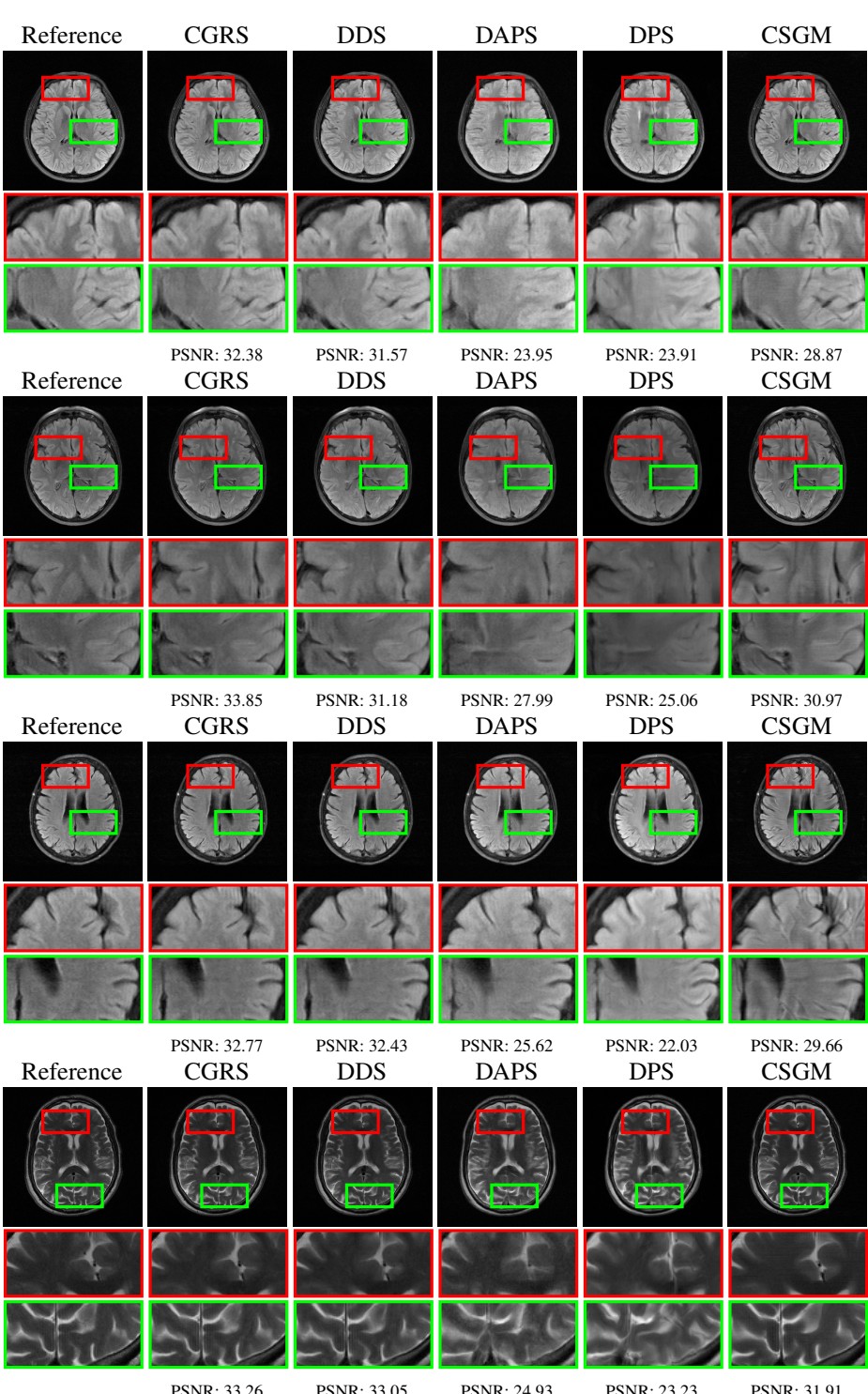

Figure 4: **More samples for Brain MRI reconstruction under ×8 acceleration**. CGRS is able to generate high-fidelity reconstructions for MRI tasks.

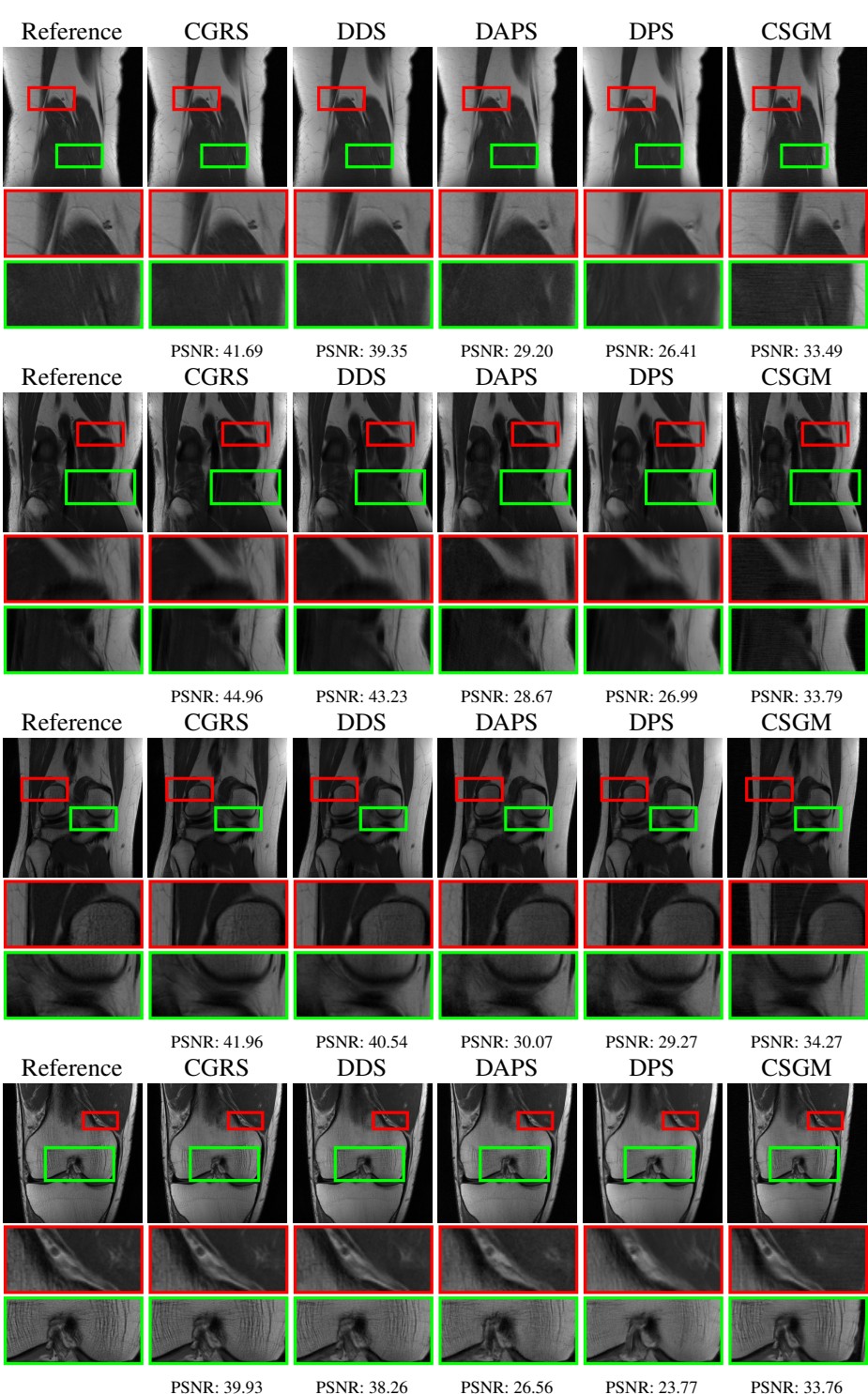

Figure 5: **More samples for Knee MRI reconstruction under ×4 acceleration**. CGRS is able to generate high-fidelity reconstructions for MRI tasks.

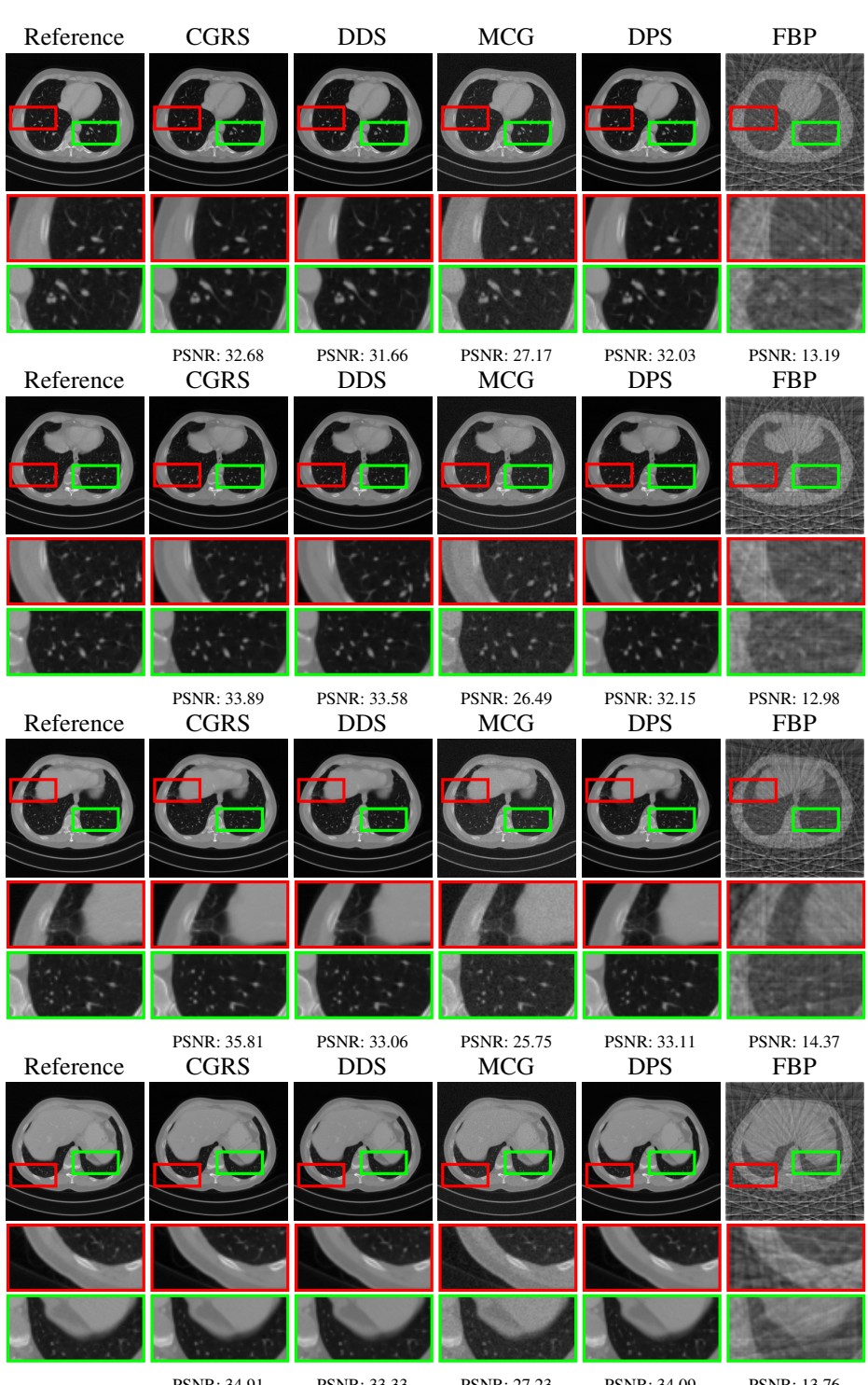

Figure 6: **More samples for CT reconstruction using 18 projection angles**.

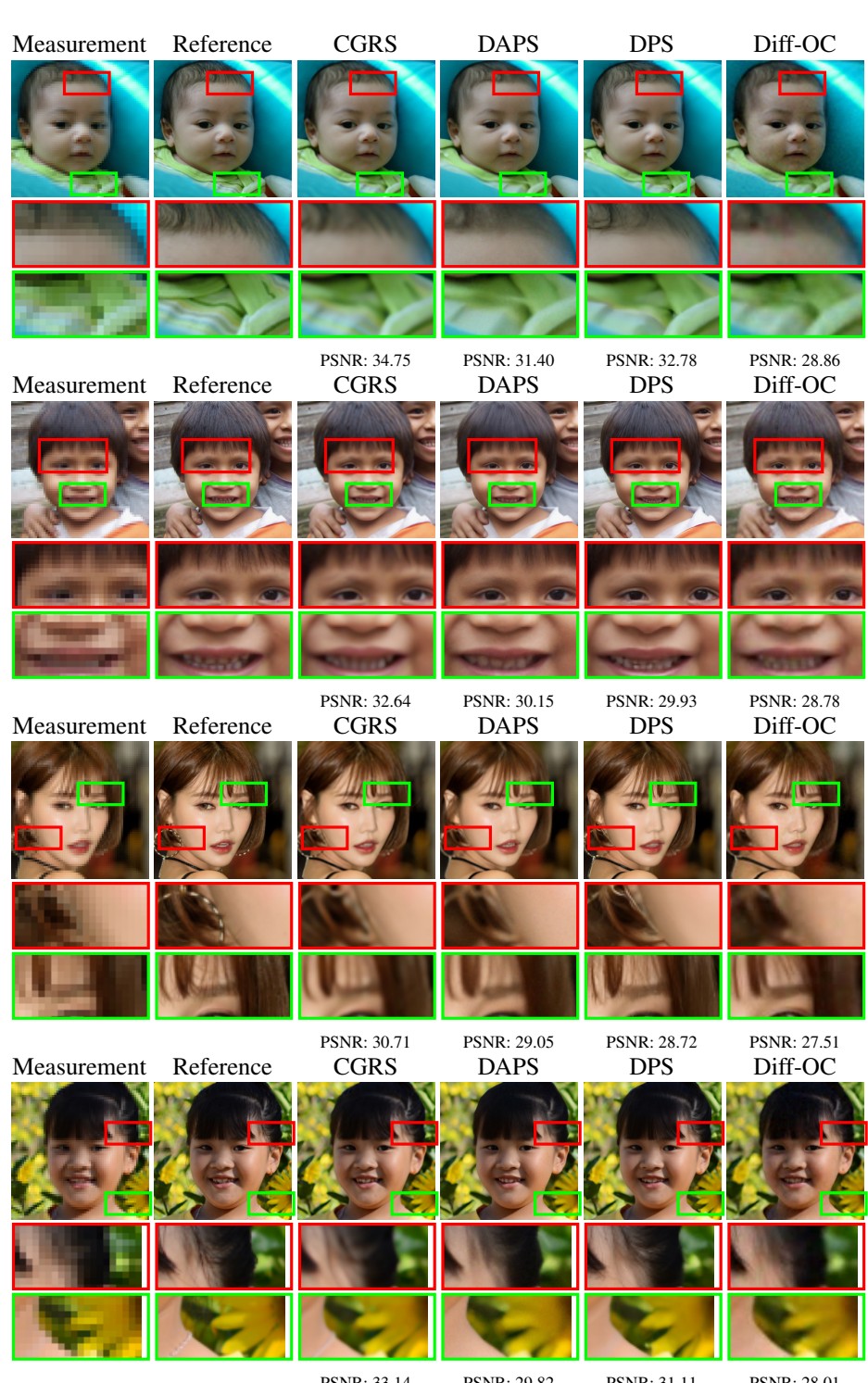

Figure 7: **Qualitative results of super-resolution ×4 on FFHQ.**

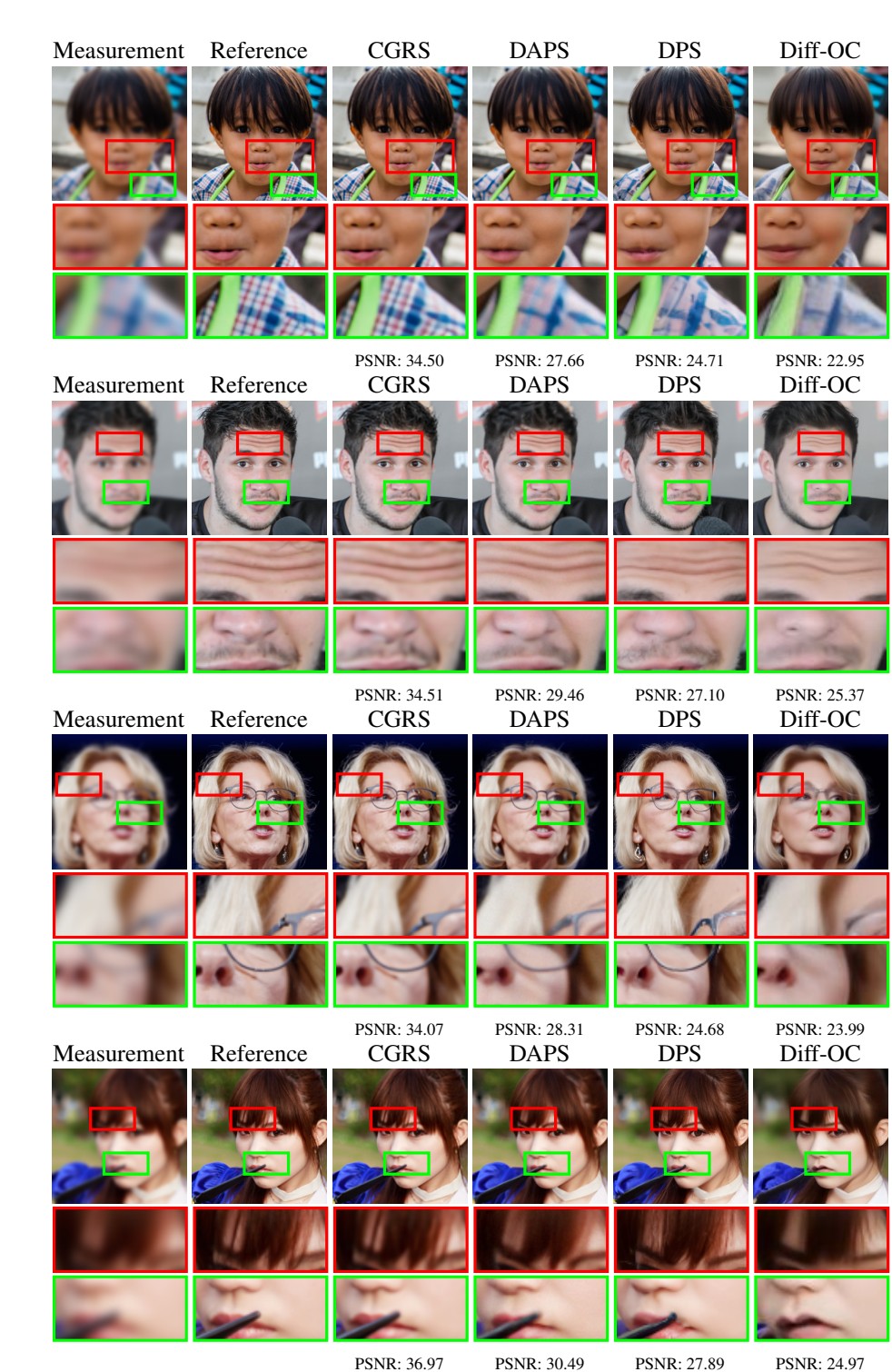

Figure 8: **Qualitative results of Gaussian deblurring on FFHQ**.

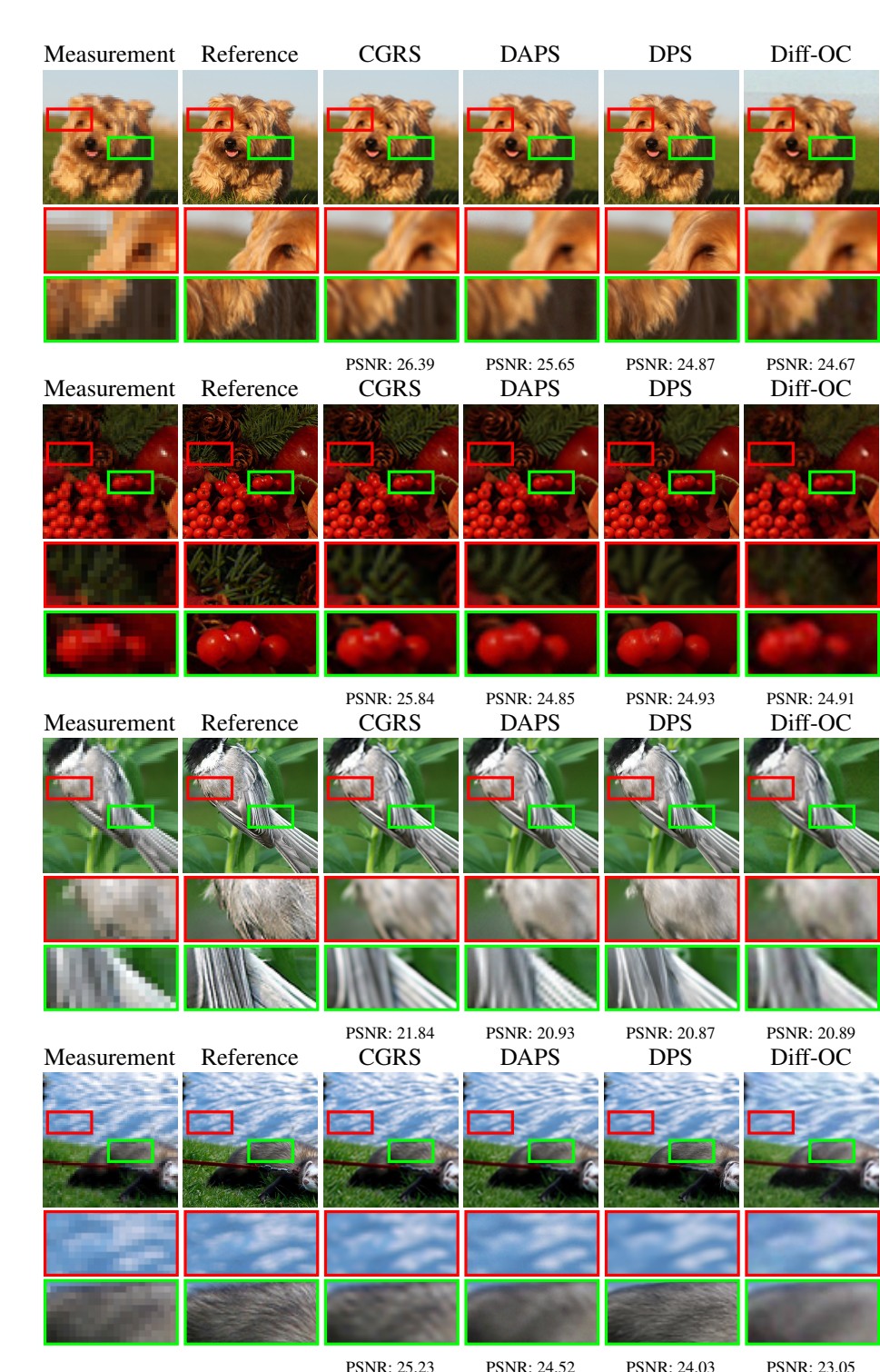

Figure 9: **Qualitative results of super-resolution ×4 on ImageNet.**

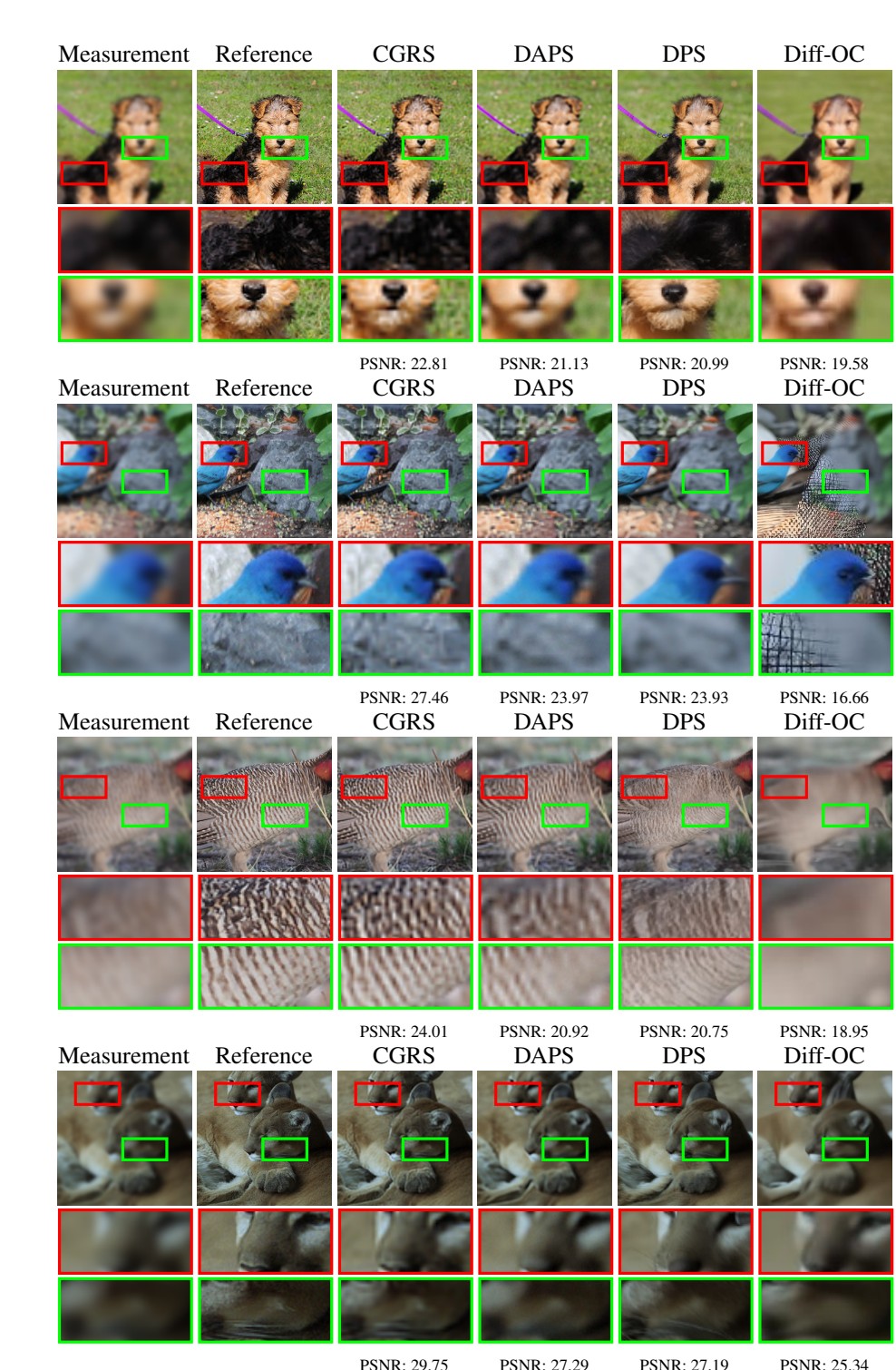

Figure 10: **Qualitative results of Gaussian deblurring on ImageNet**.

