# OpenReview forum: "Consistency-Guided Reverse Sampling for General Linear Inverse Problems"
_ICLR.cc/2026/Conference — Submitted to ICLR 2026_

### Official Review · Reviewer_R6EE · 2025-10-15

**Soundness:** 2
**Presentation:** 1
**Contribution:** 1
**Rating:** 2
**Confidence:** 5

**Summary:**

The authors propose consistency-guided reverse sampling (CGSR), as a method for solving inverse problems with diffusion models. The algorithm follows that of the usual guidance based methods, but instead of a single gradient descent step, the authors solve a minimization problem, similar to DDS [1] and DiffPIR [2].


**References**

[1] Chung, Hyungjin, Suhyeon Lee, and Jong Chul Ye. "Decomposed Diffusion Sampler for Accelerating Large-Scale Inverse Problems." ICLR 2025.

[2] Zhu, Yuanzhi, et al. "Denoising diffusion models for plug-and-play image restoration." CVPRW, 2023.

**Strengths:**

The method seems to outperform baselines.

**Weaknesses:**

Most components stated in this paper, including the use of optimization instead of gradient descent step [1,2], use of CCDF [3], etc., are already widely adopted in the diffusion models for inverse problems literature. I cannot find a contribution to the field.

**References**

[1] Chung, Hyungjin, Suhyeon Lee, and Jong Chul Ye. "Decomposed Diffusion Sampler for Accelerating Large-Scale Inverse Problems." ICLR 2025.

[2] Zhu, Yuanzhi, et al. "Denoising diffusion models for plug-and-play image restoration." CVPRW, 2023.

[3] Chung, Hyungjin, Byeongsu Sim, and Jong Chul Ye. "Come-closer-diffuse-faster: Accelerating conditional diffusion models for inverse problems through stochastic contraction." CVPR 2022.

**Questions:**

Please see weaknesses

---

### Official Review · Reviewer_6P8D · 2025-10-28

**Soundness:** 2
**Presentation:** 3
**Contribution:** 2
**Rating:** 4
**Confidence:** 4

**Summary:**

This paper proposes Consistency-Guided Reverse Sampling (CGRS) for solving general linear inverse problems, which integrates measurement-consistency optimization into each reverse diffusion step. Instead of relying solely on score function, CGRS refines each intermediate denoised estimate via a least-squares projection. Additionally, Fast CGRS is introduced, which initializes the reverse process from a coarse optimization-based reconstruction, reducing the number of diffusion steps. Experiments show effectiveness.

**Strengths:**

1. Unlike DPS or DAPS that modify the score function, CGRS operates as a projection step, conceptually closer to proximal optimization and consistent sampling.
2. The paper reports large gains in PSNR and SSIM on MRI and CT tasks (e.g., +2–3 dB improvement over DDS), and generalizes across very different problem domains (MRI, CT, natural images).

**Weaknesses:**

1. The innovation of this work is somewhat minor, and the setting is relative simple, while extension to nonlinear or non-differentiable measurement operators is unclear;
2. The proposed optimization step (Eq. 19–20) effectively turns each reverse update into a point estimate projection, which may collapse posterior diversity. It’s uncertain whether the method still approximates the true posterior or simply produces a MAP-like deterministic reconstruction.
3. The paper could include more analysis on sensitivity to noise level β_y, solver choice (CG vs ADMM), or regularization terms in the optimization.
4. Solving a least-squares problem per step adds heavy computation, especially for 1000 denoising steps, and the runtime comparisons or GPU memory profiling against other diffusion solvers are needed.
5. It would strengthen credibility to show where CGRS fails.

**Questions:**

1. Whether the authors test CGRS with latent diffusion models instead of pixel-space DMs?
2. Would combining CGRS with distillation-based fast samplers (e.g., DPM-Solver++) further reduce sampling cost?
3. Is there any instability or divergence observed when enforcing strong consistency at early timesteps?

---

### Official Review · Reviewer_USyv · 2025-10-29

**Soundness:** 1
**Presentation:** 3
**Contribution:** 2
**Rating:** 2
**Confidence:** 4

**Summary:**

The paper proposes a diffusion model-based inverse problem solver, where the measurement-consistency update is defined as a linear least-squares problem for denoised estimations. The major difference from previous methods is that it approximates $q_t(x_{t-1}|x_t, \hat x_0)$ by $q_t(x_{t-1}|\hat x_0)$, which corresponds to a forward diffusion process. Across various linear inverse problems, the proposed method consistently outperforms baseline models.

**Strengths:**

- The paper is written clearly.
- The proposed method is compared across natural images and medical images, emphasizing the effectiveness of the proposed method.

**Weaknesses:**

- Eq. (14)–(21) are already well-established equations in the related works [1,2,3]. The paper fails to reference them properly and does not provide new insights on this aspect.

- The integration term in Eq. (13) disappears in Eq. (14) without a clear justification.

- Line 761: In the proof of Proposition 2, which claims the validity of estimating $q_t(x_{t-1}|x_t, \hat x_0)$ with $q_t(x_{t-1}|\hat x_0)$, the paper simply assumes that $\hat x_0$ is independent of $x_t$ if $\hat x_0$ is the optimal solution of a linear least-squares problem. Suppose $A$ denotes a masking operator with inpainting mask $M$, represented as element-wise multiplication. Then, the linear least-squares problem has a closed-form solution $(1-M)\odot \hat x_0 + M\odot y$, where $\odot$ denotes element-wise multiplication. However, by the definition of $\hat x_0$ in Proposition 1, it cannot be independent of $x_t$. Therefore, the statement in line 761 may be incorrect, and Proposition 2 does not hold.

- Beyond the invalidity of Proposition 2, the proposed method effectively reduces to a special case of DDS with fully stochastic sampling, which substantially limits the novelty of the approach.

Reference

[1] Decomposed Diffusion Sampler for Accelerating Large-Scale Inverse Problems, ICLR 2024

[2] DreamSampler: Unifying Diffusion Sampling and Score Distillation for Image Manipulation, ECCV 2024

[3] Regularization by Texts for Latent Diffusion Inverse Solvers, ICLR 2025

**Questions:**

- Can authors provide more valid proof for the proposition 2? For example, if we follows the proposed method, it is not a generative trajectory anymore. How diffusion model prior is used while we did not follow the diffusion sampling trajectory?

- When eq (41) goes to eq (45), where is the integral? Even though we discard $x_t$ from $q(x_{t-1}|x_t, x_0)$, eq (45) should be $\int q(x_{t-1}| \hat x_0)q(\hat x_0) d\hat x_0$. It seems like there is an error due to a typo in eq (41) where we are integrating over $x_0$, not $x_t$.

---

### Official Review · Reviewer_VkQZ · 2025-11-01

**Soundness:** 3
**Presentation:** 3
**Contribution:** 2
**Rating:** 4
**Confidence:** 4

**Summary:**

This paper presents a linear inverse problem solver (CGRS) with pretrained diffusion priors. CGRS solves inverse problems by iterating among (i) partial unconditional sampling, (ii) a measurement-consistency update, and (iii) re-injecting Gaussian noise, while annealing the noise level to zero. Moreover, the authors propose to accelerate CGRS by adopting a pseudo-inverse initialization. Experiments in compressed sensing MRI, CT reconstruction, and natural image restoration demonstrate advantages of CGRS over existing methods.

**Strengths:**

- This paper is in general well-written, with well-explained theoretical results.
- Experimental results show improvement over previous methods with a clear margin.

**Weaknesses:**

- The manuscript does not adequately situate CGRS within prior diffusion-based inverse-problem methods such as DiffPIR [1], DCDP [2], DAPS [3], and SITCOM [4]. These works also employ “denoising–optimizing–noising” iterations, which the paper presents as novel. In its current form, the conceptual contribution appears incremental as it is difficult to distinguish meaningfully from DCDP [2], and the theoretical propositions largely overlap with those in DAPS [3]. This omission undermines the claimed originality and makes the conceptual advance unclear.
- All experiments use a simulated, noiseless inverse-problem setup in which measurements y are generated from a forward model applied to ground-truth images. This setting does not reflect practical data imperfections and may overstate performance. More realistic evaluation is needed, e.g., reconstructions from raw k-space (fastMRI) or with additional Gaussian measurement noise.
- This paper is missing comparison to several diffusion inverse problem solvers [5-6], with DDNM [5] specifically designed for linear forward functions.

[1] Zhu et al. "Denoising Diffusion Models for Plug-and-Play Image Restoration", CVPR 2023.

[2] Li et al. "Decoupled Data Consistency with Diffusion Purification for Image Restoration", arXiv:2403.06054.

[3] Zhang et al. "Improving Diffusion Inverse Problem Solving with Decoupled Noise Annealing", CVPR 2025.

[4] Alkhouri et al. "SITCOM: Step-wise Triple-Consistent Diffusion Sampling for Inverse Problems", ICML 2025.

[5] Wang et al.  Zero-shot image restoration using denoising diffusion null-space model. ICLR 2023.

[6] Mardani et al. "A Variational Perspective on Solving Inverse Problems with Diffusion Models", ICLR 2024.

**Questions:**

- What is the main conceptual difference of the proposed CGRS from existing methods like DCDP[2] and DAPS[3]?
- How important it is to initialize the proposed algorithm with $\arg\min_x ||y - A(x)||^2$? Is it expected to be the main reason why CGRS outperforms the considered baselines?

---

### Meta-Review · Area_Chair_B8cu · 2026-01-05

**Summary:**

The paper proposes a diffusion model-based linear inverse problem solver, which integrates measurement-consistency optimization into the reverse diffusion step. The reviewers raise the concerns about the paper contributions compared with previous literature, experiment setting, results comparison, unclear technical details in the method that may require more clarification, etc. As there is no rebuttal response and discussion to address those questions and concerns, this would be a clear rejection.

**Reviewer Concerns:**

There is no rebuttal and discussion.

**Reviewer Scores:**

There is no rebuttal and discussion.

---

### Decision · Program_Chairs · 2026-01-26

Reject